# Bioaffinity-based surface immobilization of antibodies to capture endothelial colony-forming cells

Mariève D. Boulanger[1‡], Hugo A. Level[1‡], Mohamed A. Elkhodiry[1‡], Omar S. Bashth[1], Pascale Chevallier[2,3], Gaétan Laroche[2,3], Corinne A. Hoesli[1]*

1 Department of Chemical Engineering, McGill University, Montreal, Canada, 2 Laboratoire d'Ingénierie de Surface, Centre de Recherche sur les Matériaux Avancés, Département de Génie des Mines, de la Métallurgie et des Matériaux, Université Laval, Québec, Canada, 3 Centre de Recherche du Centre Hospitalier Universitaire de Québec, Hôpital St-François d'Assise, Québec, Canada

‡ MDB, HAL and MAE are co-first authors on this work.
* corinne.hoesli@mcgill.ca, corinne.hoesli@gmail.com

**Data Availability Statement:** All relevant data are within the manuscript and its Supporting information files.

## Abstract

Maximizing the re-endothelialization of vascular implants such as prostheses or stents has the potential to significantly improve their long-term performance. Endothelial progenitor cell capture stents with surface-immobilized antibodies show significantly improved endothelialization in the clinic. However, most current antibody-based stent surface modification strategies rely on antibody adsorption or direct conjugation via amino or carboxyl groups which leads to poor control over antibody surface concentration and/or molecular orientation, and ultimately bioavailability for cell capture. Here, we assess the utility of a bioaffinity-based surface modification strategy to immobilize antibodies targeting endothelial cell surface antigens. A cysteine-tagged truncated protein G polypeptide containing three Fc-binding domains was conjugated onto aminated polystyrene substrates via a bi-functional linking arm, followed by antibody immobilization. Different IgG antibodies were successfully immobilized on the protein G-modified surfaces. Covalent grafting of the protein G polypeptide was more effective than surface adsorption in immobilizing antibodies at high density based on fluorophore-labeled secondary antibody detection, as well as endothelial colony-forming cell capture through anti-CD144 antibodies. This work presents a potential avenue for enhancing the performance of cell capture strategies by using covalent grafting of protein G polypeptides to immobilize IgG antibodies.

## 1. Introduction

Despite decades of continuous improvement in the overall performance of vascular implants, biocompatibility challenges remain a major area of concern [1, 2]. A healthy endothelium can modulate protein deposition, platelet activation, and proliferation of the underlying smooth muscle layer which are critical in reducing the risk of re-stenosis and thrombosis [3]. A promising approach to enhance stent endothelialization and reduce the risks of implant failure is to capture circulating endothelial progenitor cells (EPCs) [4]. Capturing EPCs, particularly their

**Funding:** This study was financially supported by the Canadian Institutes of Health Research (CIHR, MOP 142285; CAH and GL) and the Canadian Foundation for Innovation (CFI, project 35507; CAH). This research was undertaken, in part, thanks to funding from the Canada Research Chairs Program (CAH). This work was supported via travel awards and networking opportunities offered by ThéCell (The Quebec Network for Cell, Tissue and Gene Therapy; MDB, MAE, OSB, CH), the Quebec Center for Advanced Materials (QCAM; MDB, MAE, OSB, GL, CH), PROTEO (The Quebec Network for Research on Protein Function; MDB, MAE, OSB, CH), CMDO (the Cardiometabolic Health, Diabetes, and Obesity Research Network; MDB, MAE, OSB, CH), and the MRM (McGill Regenerative Medicine; MDB, MAE, OSB, CH) network. MDB received a Graduate Excellence Fellowship from the Faculty of Engineering of McGill University. MAE received a scholarship from the Fonds de recherche du Québec - Nature et technologies (Programme de bourses d'excellence pour étudiants étrangers 262907). OSB received a 2017 Hadhramout Establishment for Human Development scholarship. The funders had no role in study design, data collection and analysis, decision to publish, or preparation of the manuscript.

**Competing interests:** The authors have declared that no competing interests exist.

functional subtype [5] endothelial colony forming cells (ECFCs), accelerates the formation of a neo-endothelium due to their high proliferative potential and clonal expansion [6–8].

One way to promote ECFC capture on the surface of blood contacting devices is to modify the surface with antibodies that target ECFC surface antigens [9]. The surface-immobilized antibodies mimic the function of glycoproteins present on the vessel lining such as selectins and intercellular adhesion molecule-1 in the endogenous recruitment of circulating ECFCs to tissues undergoing vascular regeneration [10–12]. This antibody-based strategy was commercially adopted to create the Genous™ stent (Orbusneich, USA), an anti-CD34 antibody surface-modified metal stent. In a pilot study with 193 patients, the Genous™ stent was shown to be as safe as drug-eluting stents (the control group) with no observed difference in adverse cardiac effects. The study also revealed a promising reduction in the rate of in-stent thrombosis in patients with the Genous™ stent compared to the drug-eluting stent group despite an increased rate of re-stenosis [13].

Since the first generation of EPC capture stents, potential areas of improvement in stent design were investigated. For example, the choice of CD34 as a target antigen has been under scrutiny due to its presence on the surface of hematopoietic progenitors that can exacerbate intimal hyperplasia [14]. Stents modified with anti-vascular endothelial cadherin (CD144) were shown to be more effective in accelerating endothelialization in animal models [15]. Furthermore, available antibody-modified implants mostly utilize passive adsorption or covalent conjugation to immobilize antibodies on the surface. Using passive adsorption, non-covalent interactions between the antibody and the surface dictates the strength and longevity of the surface modification often leading to lack of durability and lack of control over antibody orientation [16, 17]. Using covalent conjugation, most strategies exploit reactive groups introduced on surfaces to conjugate antibodies via free amine or carboxyl groups present in the antibody sequence [18, 19]. The abundance of these functional groups per antibody results in random antibody orientations on the surface and could lead to changes in conformation affecting its antigen binding efficacy [20, 21]. Unwanted reactions between amino acids in the hypervariable region of the antibody and the cross-linking reagents used for covalent immobilization could also directly affect the antigen-binding capacity of the antibody.

More recently, improved bio-affinity-based antibody immobilization techniques emerged as an alternative that can enhance the potential of next-generation ECFC-capture stents [22–24]. Li *et al.* demonstrated that stent surfaces modified with anti-CD34 antibodies immobilized via the Fc-binding protein A improved *in vivo* stent endothelialization compared to unmodified controls [25]. However, due to the layer-by-layer coating strategy applied in this study, protein A molecules can take different conformations on the surface which reduces the availability of sites available for antibody immobilization and hence antibody surface density. Recombinant DNA technology can be used to introduce cysteine residues in the sequence of Fc-binding proteins which can then be conjugated onto surfaces via the thiol group. Gold substrates modified with cysteine-tagged protein G increased antibody surface density compared to adsorption controls while maintaining optimal antibody orientation leading to enhanced antigen binding capacity. This technique has been successfully applied to design immune-assays on a chip [26, 27]. Oriented surface immobilization of antibodies via covalent grafting of cysteine-tagged protein G remains untested for *in vivo* cell capture applications, particularly of ECFCs.

Here, we grafted a cysteine-tagged protein G polypeptide containing three Fc-binding domains on polystyrene, the most commonly-used tissue culture plastic, in order to immobilize antibodies which recognize endothelial surface markers such as CD31 and CD144. We hypothesized that our modified surfaces would increase ECFC capture compared to conventional antibody immobilization via passive adsorption. To test this hypothesis, a parallel plate

flow chamber was used to study how ECFCs—derived from human donors—are captured under dynamic flow conditions.

## 2. Materials and methods

### 2.1 Surface modification and antibody immobilization

The bioaffinity-based antibody immobilization strategy relied on a 3-step process consisting of (1) activation of an aminated surface with an amine to sulfhydryl heterobifunctional linker, (2) covalent attachment of a cysteine-tagged protein G sequence containing three Fc-binding domains through sulfhydryl/maleimide reaction and (3) addition of IgG antibodies (Fig 1). First, aminated surfaces (BD PureCoat™ Amine Culture Dishes, BD Biosciences, now a part of Corning®, VWR) were reacted for 2 h with 150 μL/cm² of a 3 mg/mL suspension of sulfo-suc-cinimidyl-4-(*p*-maleimidophenyl)-butyrate (S-SMPB, #BC24, G-Biosciences) in phosphate buffered saline solution (PBS, #21600010, Thermo Fisher Scientific). Next, the cysteine-tagged protein G sequence was attached to the linking arm by adding 150 μL/cm² of a 5.5 μM recom-binant cysteine-tagged protein G polypeptide (henceforth termed "protein G polypeptide"—a recombinant non-glycosylated polypeptide chain produced in *E. coli* containing an N-terminal Cys followed by amino acids 298–497 of the streptococcal protein G sequence, #PRO-1328, Prospec-Tany Technogene Ltd) solution in PBS for 1 h. Finally, primary antibodies targeting cell surface antigens (mouse anti-human CD31 antibody #303101; mouse anti-human CD105 #323202; mouse anti-human CD144 #348502; and mouse anti-human CD14, anti-CD14, #367102; all from Biolegend, San Diego, US) were immobilized on the protein-G modified surfaces by adding 150 μL/cm² of antibody solution at 5 μg/mL in PBS for 1 h. Surfaces were then rinsed twice with PBS, once with a 1% SDS-TRIS pH 11 solution (5% v/v of 20% sodium dodecyl sulfate #05030, from Sigma Aldrich and 2.4% w/v TRIS base PBP151-500 from Fisher Scientific in reverse osmosis water, pH adjusted to 11 with 2N NaOH solution) to remove adsorbed molecules, twice with PBS, and finally rinsed with reverse osmosis (RO) filtered water. The surfaces were then air-dried and stored for at most 1 week at room temperature before use. Adsorption controls followed the same surface modification scheme, except for the omission of S-SMPB prior to incubation with cysteine-tagged protein G polypeptide.

For cell capture experiments under flow, surfaces were obtained by cutting aminated poly-styrene Petri dishes (BD Purecoat™ Amine #354732, BD Biosciences, San Jose, USA) into 3.0 cm × 2.5 cm slides using a Micro Mill (Datron Neo 3-axis CNC Mill, Cell imaging and analysis

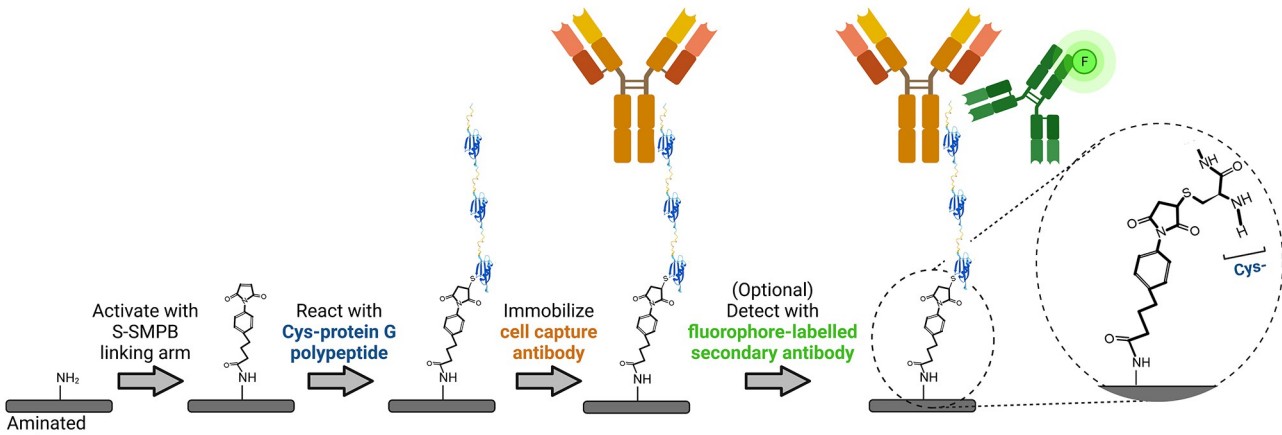

**Fig 1. Schematic representation of the antibody immobilization process followed by a fluorescence-based antibody detection step.**

network, McGill University, Canada). The circumference of these cut samples was lined with Teflon™ tape (#3213–103, polytetrafluoroethylene (PTFE) thread sealant) to maintain solutions on surfaces during the reaction steps. All other surface modifications were performed directly in well plates. All incubation steps were performed in the dark with 90 rpm agitation on a rotary shaker (Ecotron, Infors HT) at room temperature. After each reaction step, solutions containing reactants were removed, and surfaces were rinsed twice with 0.2 μm-filtered PBS. Collagen-coated surfaces were used as a native protein control and were prepared by adding 0.15 mL/cm$^2$ of 50 μg/mL type 1 rat-tail collagen (Thermo Fisher Scientific™) in 0.02 N acetic acid (Thermo Fisher Scientific™). All surfaces were sterilized by incubation in 95% ethanol.

## 2.2 X-Ray Photoelectron Spectroscopy (XPS)

The chemical composition of the aminated surfaces, before and after functionalization was investigated by XPS using a PHI 5600-ci spectrometer (Physical Electronics, Eden Prairie, MN). The main XPS chamber was maintained at a base pressure of $< 8{\times}10^{-9}$ Torr. A standard aluminum X-ray (Al Kα = 1486.6 eV) source was used at 300 W to record survey spectra with charge neutralization, while C1s high resolution spectra were recorded with a standard magnesium X-ray source without neutralization. The detection angle was set at 45º with respect to the normal of the surface and the analyzed area was 0.5 mm$^2$. The spectrometer work function was adjusted to 285.0 eV for the main C (1s) peak. Curve fitting of high-resolution peaks were determined by means of the least squares minimization procedure employing Gaussian-Lorentzian functions and a Shirley-type background.

## 2.3 Enzyme-Linked Immunosorbent Assay (ELISA) for protein G

A direct ELISA was developed to detect and quantify protein G polypeptide surface concentrations. Purecoat™ 96 well plates (BD Purecoat™ Amine # 356717, BD Biosciences) were functionalized as described above (S-SMPB(+)) or by omitting the S-SMPB activation step (S-SMPB(-)). To block further protein adsorption, 200 μL/well of 1% BSA solution in PBS was introduced and left to incubate for 90 min at 37ºC on a rotary shaker at 90 RPM. Wells were rinsed twice with wash buffer consisting of 0.05% Tween-20 (#P1379, Sigma-Aldrich) solution in PBS. To detect protein G polypeptide, a chicken immunoglobulin Y (IgY) anti-protein G was used. This antibody was selected due to the absence of affinity between protein G and the Fc fragment of IgY antibodies [28]: only the antigen binding fragment of the IgY anti-protein G can interact with protein G, which should facilitate quantification of surface ligands [29]. A volume of 100 μL of horseradish peroxidase (HRP)-conjugated IgY anti-protein G secondary antibody (HRP anti-protein G, OAIA00498, Aviva systems biology) solution (0.02 μg/mL of HRP anti-protein G diluted in rinsing solution with 1% BSA) was incubated in each well for 2 h at room temperature. Wells were immediately rinsed once with 1% SDS-TRIS solution at pH 11 and twice with wash buffer. To detect HRP, 100 μL of Slow TMB-ELISA substrate solution (#34024, Thermo Fisher) was added per well. After 25 min of incubation without agitation at room temperature, 100 μL/well of 1M sulfuric acid solution was added to stop the reaction, and absorbance measurements were immediately taken at 450 nm on a Benchmark™ plate reader (Bio-Rad, Berkeley, USA).

## 2.4 Immobilized antibody detection and quantification

Purecoat™ amine surfaces were modified as described above, but only certain regions of test surfaces were treated with protein G polypeptide by adding spots of 0.5 μL cysteine-tagged protein G polypeptide solution at concentrations ranging between 0.055 μM and 55 μM. To assess the effect of adsorption on surface amounts of protein G polypeptide, the spots were

deposited on surfaces with (S-SMPB (+)) or without (S-SMPB (-)) S-SMPB treatment (previously treated with 3 mg/mL for 2 hours unless otherwise mentioned). After 1 h incubation, surfaces were rinsed with PBS and covered (spot and surrounding region) with primary antibody solution for 1 h as described above. After two washes in PBS, surfaces were covered with Alexa Fluor 488-conjugated goat anti-mouse secondary antibody solution at 20 μg/mL. After 1 h of incubation at room temperature, surfaces were rinsed twice with 1% SDS-TRIS solution, twice with PBS and twice with RO water before air drying. Spots were then imaged using a laser scanning confocal microscope (Zeiss LSM 5 Exciter, Germany) at 10X with an argon laser (488 nm). A total of 10 images per spot were taken to obtain the mean fluorescence intensity of one spot along with the associated standard deviation value. A total of 3 spots per replicate were studied to obtain the mean fluorescence intensity of each condition.

## 2.5 ECFC capture under flow

Peripheral blood mononuclear cells (PBMCs) were isolated from fresh adult human peripheral blood and ECFCs were expanded as previously described [30]. Fresh blood samples were collected from adult donors (N = 4: 2 females and 2 males; mean age 25.5 years) under informed consent following a protocol (Study No. A06-M33-15A) approved by the Ethics Institutional Review Board at McGill University. To study cell capture by antibody-modified surfaces under flow, functionalized surfaces were assembled into a custom parallel-plate flow chamber system with 4 independent chambers and flow paths, as previously described [31]. The flow chamber was ethanol-sterilized and then assembled inside an incubator with humidified air maintained at 5% $CO_2$ and 37 °C. Each chamber was connected to a reservoir that was pre-filled with 15 mL of warm serum-free EGM-2 (endothelial cell growth medium-2 without serum added from the kit, Lonza). ECFCs were harvested and resuspended in serum-free EGM-2 and added to the reservoirs to reach an overall cell density of 125,000 cells/mL. A peristaltic pump (Masterflex RK-7543-02 with Masterflex L/S two channels Easy Load II pump head using L/S 13 BPT tubing) was used to circulate the cell suspension in the system at a flow rate of 0.18 mL/s to obtain 1.5 dyn/cm$^2$ wall shear stress. After 1 h of circulation, cells were fixed using a 4% paraformaldehyde solution (VWR) for 10 min, rinsed once in PBS and stored in PBS for immunocytochemistry.

## 2.6 Immunohistochemistry and microscopy

Fixed cells were permeabilized for 15 min with 0.1% Triton X (VWR) in PBS. Nuclei were stained with 1 μg/mL DAPI (Sigma) diluted in RO water for 10 min. Slides were then rinsed with RO water and stored in PBS before being imaged on an inverted fluorescent microscope (Olympus IX81). Images were acquired at 10X in phase contrast and fluorescence. At least 40 phase contrast images were acquired on each slide per cell capture experiment. Captured cells were enumerated using the "analyze particles command" of ImageJ from the 40 acquired DAPI images.

## 2.7 Statistical methods

Statistical analysis was performed with JMP Pro 13 software (SAS Institute, Cary, NC). Unless otherwise stated, data represent the average ± the standard error of the mean of 3 independent experiments. The criteria for statistically significant differences was selected to be $p < 0.05$. The Shapiro-Wilk normality test was applied prior to performing parametric tests. Student's t-test was used for comparisons between two sample groups and comparisons between multiple groups were performed using two-way analysis of variance (ANOVA) followed by Tukey-

Kramer HSD post hoc test. For ECFC capture experiments, each of the 4 replicas was conducted with ECFCs derived from a different donor.

## 3. Results

To develop a suitable antibody screening platform for cell capture, the proposed surface modification steps were first characterized, followed by testing the effect of different immobilized antibodies on ECFC capture under laminar flow.

### 3.1 Characterization of the Purecoat™ substrate and S-SMPB activation of the surface

Cysteine-tagged protein G polypeptide was conjugated onto commercially-available aminated polystyrene surfaces via the amine-to-sulfhydryl linking arm S-SMPB (Fig 1). The surface compositon at each step was assessed by XPS (Table 1). As expected, the nitrogen content decreased after S-SMPB treatment when compared to the initial aminated substrates. The surface density of amino groups also decreased after the S-SMPB activation step based on the concentration of surface-bound Orange II (Fig A1 in S1 Appendix). The carbon content increased, and oxygen content decreased after the S-SMPB reaction due to the elemental composition of the linking arm. The increase in water static contact angles observed after S-SMPB treatment (Fig A1 in S1 Appendix) is consistent with the reduction in O and N content observed by XPS and addition of hydrophobic moieties present in the S-SMPB structure. All experiments were conducted at 3 mg/mL S-SMPB in this study, but this concentration could potentially be reduced based on the Orange II and water contact angle results. Subsequent protein G polypeptide grafting led to an increase in the O and N content related to the presence of these atoms in amino acid side chains and C terminus. The penetration depth of XPS analysis is ~5 nm which is inferior to the size of antibodies and on the same range as the size of a ~22 kDa polypeptide (~2 nm expected size) such as the protein G sequence used here. Atomic ratios therefore provide a useful metric to determine whether protein G polypeptide and antibody immobilization steps were successful. The O/C and N/C ratios decreased when anti-CD144 antibodies were immobilized on the surface. This observation suggests that anti-CD144 antibodies have higher C-rich amino acid content than the protein G polypeptide. This observation was corroborated by C1s high-resolution spectra (Fig 2): the peak at 285.0 eV, associated to C-C/C-H bonds, reached 76% for CD144 surface whereas it was 65% on Cys-Protein G.

### 3.2 Analysis of protein G polypeptide grafting efficiency

Next, cysteine-tagged protein G polypeptide grafting was evaluated with a direct ELISA developed to detect and confirm protein G polypeptide presence on surfaces (Fig 3). As shown in

**Table 1. Surface atomic composition assessed by XPS survey analyses*.**

| Reaction step after which XPS analysis was conducted | Atomic percentage | | | Atomic ratios | |
| --- | --- | --- | --- | --- | --- |
| | %C | %O | %N | N/C | O/C |
| Initial pureCoat™ amine polystyrene surfaces | 63 ± 2 | 17.2 ± 0.9 | 18.8 ± 0.6 | 0.30 ± 0.02 | 0.26 ± 0.02 |
| After S-SMPB | 86 ± 1 | 7.6 ± 0.9 | 5.0 ± 0.5 | 0.057 ± 0.006 | 0.09 ± 0.01 |
| After Cys-protein G | 83.3 ± 0.9 | 9.7 ± 0.7 | 7.0 ± 0.4 | 0.084 ± 0.006 | 0.116 ± 0.009 |
| After antibody immobilization | 85.8 ± 0.9 | 8.5 ± 0.9 | 4.7 ± 1.1 | 0.05 ± 0.01 | 0.09 ± 0.01 |

* Error estimates represent the standard deviation of areas analyzed on one sample to assess the grafting homogeneity.

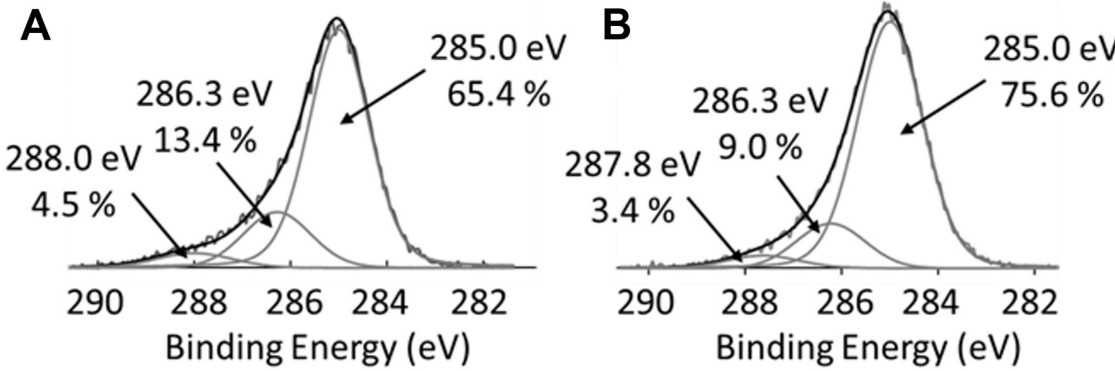

**Fig 2. High resolution C1s XPS spectra. (A)** Surfaces analyzed after the protein G polypeptide graftin step. **(B)** Surfaces after the antibody immobilization step using anti-CD144 antibodies.

Fig 3, a positive correlation between protein G polypeptide concentration and absorbance signal was observed under covalent conjugation conditions. This trend was reproduced on other commercially-available aminated polystyrene substrates (Fig A2 in S1 Appendix). This correlation was not observed for adsorbed protein G polypeptide in the absence of the linking arm. This suggests that the covalent conjugation method via S-SMPB activation improved control over the amount of protein G polypeptide present on surface compared to adsorption. Together, the results shown in Table 1, Figs 2, and 3 highlight the effectiveness of this covalent conjugation strategy in grafting protein G polypeptide in controlled amounts on amine-functionalized surfaces.

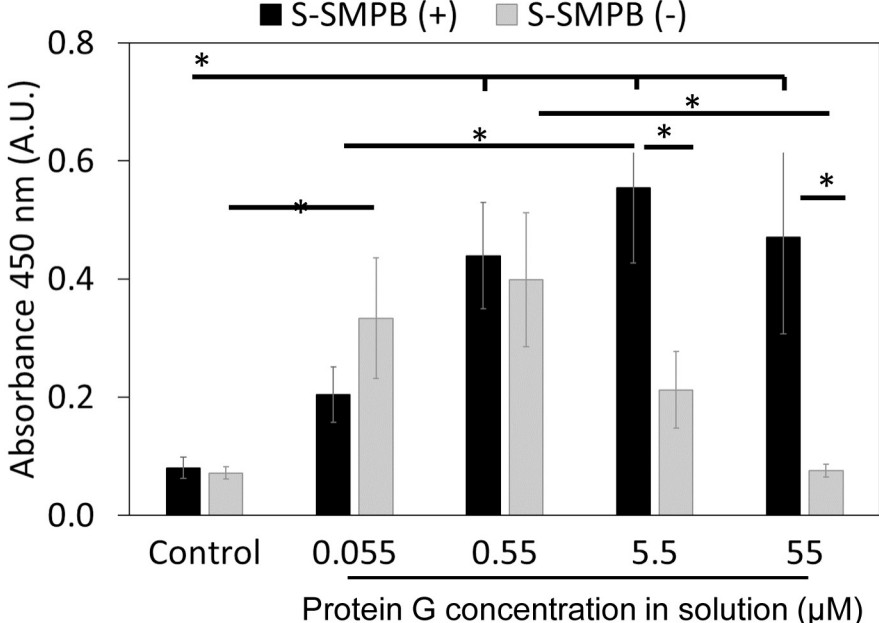

**Fig 3. Direct ELISA detection of surface-immobilized protein G polypeptide.** Conditions tested include covalent conjugation (S-SMPB(+) or adsorption (S-SMPB omitted during the first reaction step shown in Fig 1, S-SMPB(-)) of cysteine-tagged protein G polypeptide applied at different concentrations in solution. Control: no protein G polypeptide added. *P < 0.05 with N = 3. Raw data for this Fig 3 can be found in S2 Appendix.

### 3.3 Antibodies interact specifically with protein G modified surfaces

After covalent conjugation of protein G, the next step was to immobilize IgG antibodies onto the functionalized surfaces. As shown in Fig 4B, anti-CD31 antibodies were successfully immobilized on surfaces functionalized with protein G polypeptide based on fluorescent secondary antibody detection. The fluorescence intensity in the region where protein G polypeptide was deposited was significantly higher when applying at least 5.5 µM of cysteine-tagged

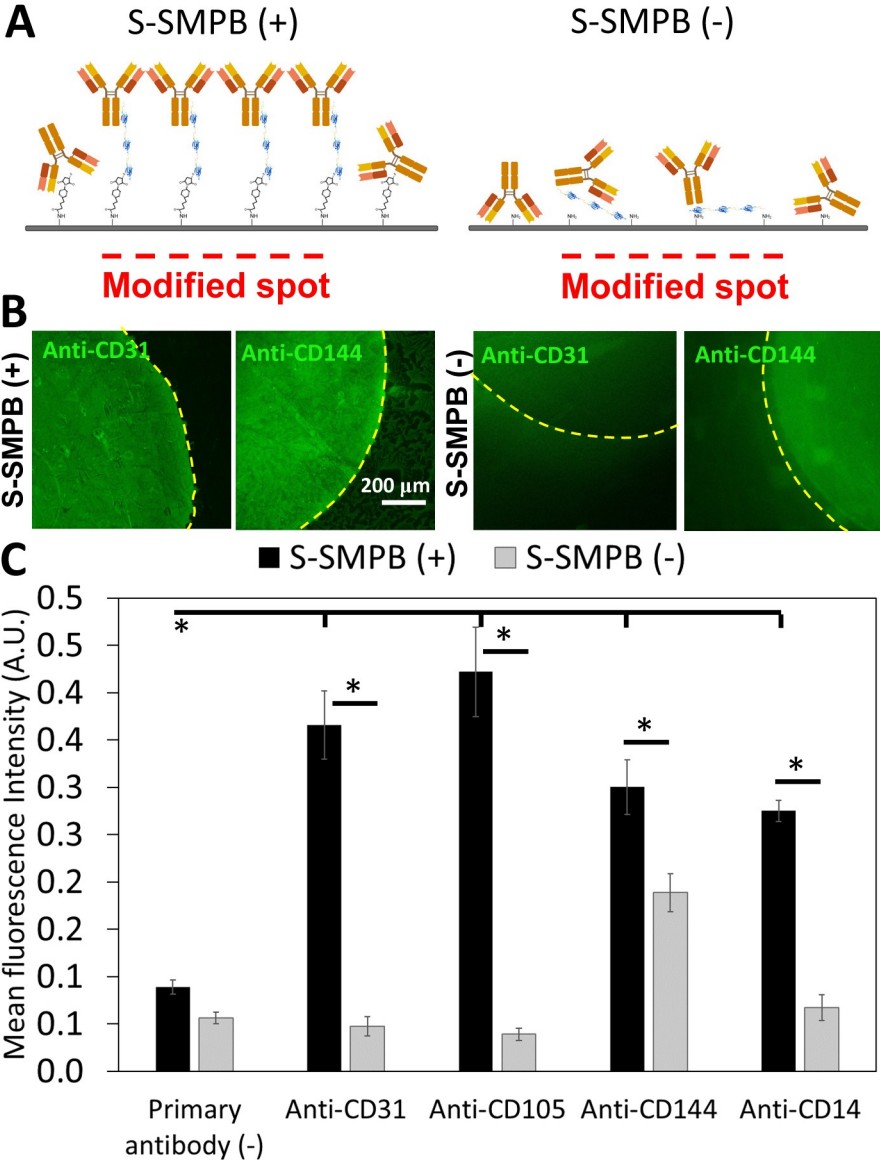

**Fig 4. Fluorescence-based detection of immobilized IgG antibodies on conjugated protein G polypeptide spots.**
**(A)** Schematic representation of antibody immobilization on protein G spots with (S-SMPB(+)) or without (S-SMPB (-)) covalent grafting of protein G polypeptide. **(B)** Spotted anti-CD31 or anti-CD144 antibodies detected through fluorophore-labelled anti-mouse antibodies as described in the last step of Fig 1. The protein G polypeptide concentration applied was 5.5 µM. **(C)** Successful immobilization of different primary antibodies on conjugated protein G polypeptide (S-SMPB (+)) based on the detection of fluorophore-labeled secondary antibodies added after protein G polypeptide and primary antibody immobilization. Surfaces without S-SMPB and/or without primary antibodies were used as negative controls. *P < 0.05 with N = 3. Raw data for Fig 4C can be found in S3 Appendix.

protein G polypeptide as compared to 0.55 μM or to the surrounding region without polypeptide (Fig A3 in S1 Appendix). This was not observed on surfaces with adsorbed protein G polypeptide (S-SMPB(-)). A concentration 5.5 μM of protein G was selected for further studies given the higher background signal observed when applying 55 μM protein G polypeptide on S-SMPB(-) surfaces (Fig A3 in S1 Appendix). These experiments were repeated with four different IgG antibodies targeting endothelial (anti-CD31, anti-CD105, anti-CD144) or macrophage/monocyte (anti-CD14) surface markers which were successfully immobilized on surfaces using the same strategy (Fig 4C). Atomic force microscopy imaging performed on the modified surfaces also shows high density of immobilized antibodies. Immobilization via the protein G polypeptide created features of 10 nm ~ 15 nm height on surfaces (Fig A4 in S1 Appendix), as would be expected for oriented antibodies immobilized via Fc domains [26].

### 3.4 Antibody-functionalized surfaces can capture ECFCs

ECFCs were injected into a flow loop with a parallel plate chamber and circulated for 1 h at 1.5 dyn/cm$^2$ wall-shear stress to determine whether antibody-modified surfaces can mediate cell capture. This wall shear stress is at the lower end of the physiological range and was selected to allow quantification of cell capture *in vitro* [32, 33]. Surfaces with (1) adsorbed anti-CD144 (S-SMPB (-)), (2) collagen-coated surfaces, (3) immobilized anti-CD14 on conjugated protein G polypeptide (S-SMPB (+)) and (4) immobilized anti-CD144 on conjugated protein G polypeptide (S-SMPB (+)) were tested in the flow system. Out of the four conditions, only the immobilized anti-CD144 on the conjugated protein G polypeptide had a significant effect in enhancing ECFC capture under flow (Fig 5). Surfaces with adsorbed anti-CD144 showed no significant difference in the number of captured ECFCs compared with surfaces modified with anti-CD14, a surface antigen that is not expressed by ECFCs [30]. Surfaces coated with rat tail collagen, a commonly used ECFC substrate, also had a significantly lower number of captured cells compared to the immobilized anti-CD144 on the conjugated protein G polypeptide. As a control, PBMCs, rich in CD14+ cells (>45%) but with low or undetectable CD144+ cell populations (<0.1%), were separately circulated in the same conditions over the same surfaces. In this arrangement, a significantly higher number of captured cells were observed on surfaces with immobilized anti-CD14 compared to surfaces with anti-CD144, confirming the specificity of the cell capturing strategy (S1 Appendix).

## 4. Discussion

To our knowledge, this study is the first demonstration of the utility of the covalent conjugation of polypeptides containing the Fc-binding domains of protein G for antibody immobilization towards cell capture applications. This strategy was selected to maximize the immunoaffinity of the antibodies compared to adsorption or direct covalent conjugation methods which can result in random orientation and reduced availability of antigen-binding sites [34–36]. The mild conditions (physiological pH, room temperature, aqueous conditions) of this surface modification strategy assures compatibility with a wide variety of cell culture substrates and biomaterials. The covalent conjugation of the protein G polypeptide via its N-terminal cysteine tag enhances the ability to control the orientation of the protein G and of the immobilized antibody [26]. Different IgG antibodies were successfully immobilized on conjugated protein G polypeptide, achieving better control over protein G polypeptide surface density compared with adsorption (omission of the linking arm). The surfaces with grafted protein G polypeptide and immobilized anti-CD144 successfully captured circulating ECFCs at 1.5 dyn/cm$^2$ wall shear stress, contrary to surfaces where the linking arm was omitted from the surface treatment. The grafted protein G polypeptide anti-CD144 surfaces also captured

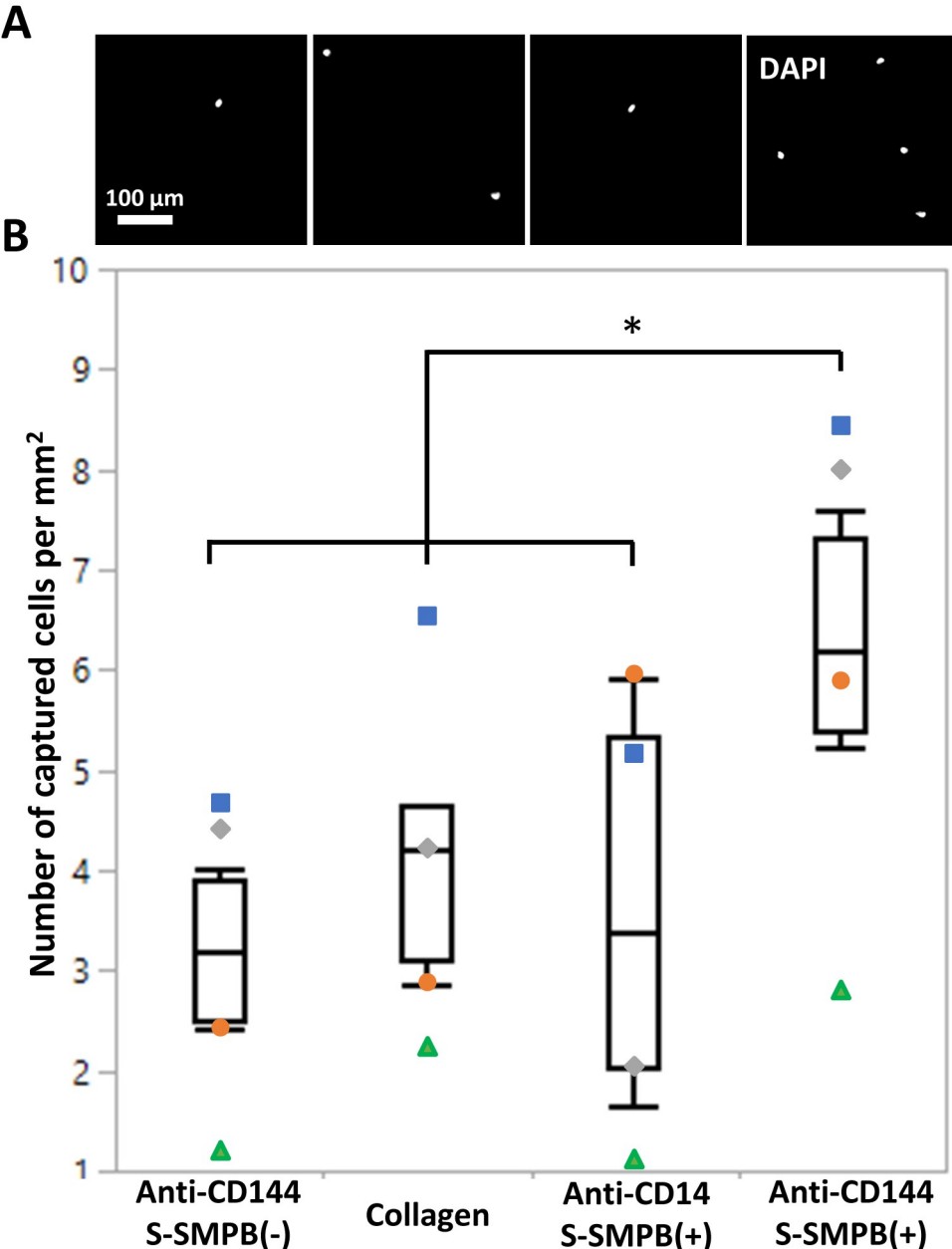

**Fig 5. ECFC capture from laminar flow conditions on functionalized surfaces.** Conditions tested include anti-C144 (present on ECFCs) on adsorbed or conjugated protein G polypeptide, anti-CD14 (not present on ECFCs) on conjugated protein G polypeptide (negative control), or collagen. **(A)** Fluorescence images of ECFC nuclei on modified surfaces after 1 h of exposure to cell suspension under flow conditions. **(B)** Quantification of number of cells per mm$^2$ on the modified surfaces at the end of the 1 h of flow. Each symbol represents data collected using ECFCs from a separate donor. $^*$P $<$ 0.05 with N = 4.

significantly higher circulating ECFC numbers compared with surfaces with antibodies which do not target ECFCs (anti-CD14). Compared to a native extracellular matrix protein such as collagen, the anti-CD144 antibodies immobilized via grafted protein G polypeptide resulted in significantly higher levels of ECFC capture demonstrating the advantage of targeting specific cell-surface antigens. These promising findings highlight the value of our proposed surface modification strategy for the design of EPC capture vascular biomaterials.

S-SMPB is a versatile linking arm which has been applied to vascular biomaterials such as aminated polytetrafluoroethylene (PTFE) [37], poly (L-lactide (PLLA), poly (ε-caprolactone) (PCL)[38] and other aminated model surfaces [39]. The maleimide functional group of the S-SMPB reacts with the sulfhydryl group present solely in the cysteine tag of the protein G polypeptide, thus creating a selective oriented conjugation strategy. Compared to protein G polypeptide adsorption, conjugation via S-SMPB led to a better control over protein G polypeptide surface density. With covalent protein G polypeptide grafting, surface concentration of protein G polypeptide followed an expected saturation profile (Fig 3). Conversely, the maximum achievable protein G polypeptide surface concentration achieved through adsorption was lower and was followed by a decreased in surface densities at higher concentrations. A possible explanation for the adsorption profiles at high polypeptide concentrations in solution is the ability of free cysteines to interact in solution to form disulfide bonds which can produce dimers reducing effective interaction with the detection antibodies of the ELISA. High concentrations may also lead to multilayers on surfaces with changes in conformation leading to reduced secondary antibody detection. Therefore, the conjugation scheme with the S-SMPB linking arm allowed better control over protein G surface density.

The protein G polypeptide used in this study comprises amino acids 298–497 of the full streptococcal protein G sequence, which includes the three Fc-binding domains of this protein. Streptococcal protein G interacts with immunoglobulin G antibodies of most mammalian species [40, 41]. This interaction can be mediated both through Fc and Fab regions [42], although the association constant is one order of magnitude higher for Fc fragments [43] with the exception of mouse IgG1 where Fab plays a more significant role [44, 45]. A polypeptide fragment containing the sequence used in the current study was previously shown to interact specifically with the Fc and not the Fab [42, 46] region of human IgG antibodies. All antibodies in the current work were of IgG1 subclass except anti-CD144, of IgG2a subclass. It is possible that the difference in subclass led to different antibody interactions with adsorbed vs conjugated protein G observed in Fig 5. Theoretically, the higher ratio between Fc:Fab protein G binding affinity expected for IgG2 antibodies would lead to improved orientation on surfaces. However, even for antibodies where Fab interactions with protein G are significant, the hypervariable region is expected to remain available for antigen recognition [47].

We have recently shown that antibodies immobilized through surface-grafted RRGW peptides designed to interact with the Fc region of IgG2a antibodies can selectively capture ECFCs from a mixture of cells under dynamic flow conditions [23]. A major obstacle hampering the use of protein G on implanted biomaterials such as stents is its unknown immunogenic profile and the possibility that it can provoke undesirable host immune responses [48]. Fc-binding peptides such as RRGW, on the other hand, can pose a lower risk of triggering an immune response due to its small chemically defined structure and the absence of endotoxins due to chemical synthesis. Larger polypeptides can also be more susceptible to enzymatic degradation and thermal denaturation compared to smaller Fc-binding peptides. A potential advantage of the protein G polypeptide strategy over the previously proposed RRGW peptide are the existing protein G supply chains allowing its use in cGMP bioprocessing plants, which may facilitate large-scale use in other biomedical and clinical applications [49]. Furthermore, the recombinant cysteine-tagged protein G polypeptide has 25 times the molecular weight of the RRGW peptide which can lead to 2 to 3 nm additional spacing between the antigen-binding site and the modified surface, reducing steric hindrance [27]. There are three Fc fragment binding sites available on each protein G polypeptide compared to the RRGW peptide's single antibody binding capacity, potentially increasing the density of antibodies which can be immobilized on protein G polypeptide modified surfaces. Further

development of antibody immobilization strategies via protein G polypeptide and RRGW for *in vivo* use will require side-by-side comparison of the hemocompatibility, immunogenicity, and stability of both molecules. Given the promising results obtained via protein G-mediated antibody immobilization for cell capture, other methods that aim to orient antibodies on surfaces [50] could also show significant promise in cell capture applications. Comparison to other antibody-based biocoatings [51–53] which lead to varying degrees of antibody orientation on surfaces could help elucidate the effect of antibody conformation on cell capture efficiency.

Given the versatility of the proposed bioaffinity-based antibody immobilization strategy as demonstrated through immobilization of IgGs targeting different cell antigens, it would be interesting to study selective capture of different immune cell subsets from peripheral blood. The proliferation of ECFC post-capture should also be assessed. Combination of Fc-binding peptides or polypeptides with integrin-binding peptides shows significant promise in this regard [54]. This bi-functionnal coating could also be applied on various substrates to efficiently isolate ECFC from peripheral blood *in vitro*, paving the way to the development of novel cell-culture materials. All in all, the proposed bioaffinity-based antibody immobilization strategy shows promise towards engineering clinically successful EPC-capture biomaterials.

## 5. Conclusion

This study presents a 3-step surface functionalization strategy to immobilize antibodies on aminated surfaces via bioaffinity interactions with a protein G polypeptide containing its three Fc-binding domains. This technology can be applied to engineer endothelial progenitor cell capture stents and other cell separation devices. Model aminated polystyrene surfaces were first reacted with an amine to sulfhydryl linking arm. The linking arm was then used to conjugate protein G to the surface through a cysteine tag maximizing its antibody immobilization capacity. Different IgG antibodies were successfully immobilized on the surface and detected using a simple fluorescence-based approach. Finally, surfaces modified with anti-CD144 via our protein G polypeptide-based approach displayed superior ability in capturing human derived ECFCs from flow compared to surfaces modified through passive adsorption. Our work highlights the potential of grafted protein G-based surface functionalization strategies in enhancing the potential of ECFC capture on the surface of vascular implants. Bioaffinity-based antibody immobilization on EPC capture stents may accelerate the endothelialization process essential in vascular regeneration and homeostasis.

## Supporting information

**S1 Appendix. Supplementary methods and figures: Orange II assay, static water contact angle, atomic force microscopy and peripheral blood mononuclear cell capture on anti-CD144 and anti-CD14 surfaces.**
(PDF)

**S2 Appendix. Spreadsheet comprising the raw ELISA readouts used to generate Fig 3.**
(XLSX)

**S3 Appendix. Spreadsheet comprising the raw fluorescence intensity data used to generate Fig 4C.**
(XLSX)

## Acknowledgments

The authors thank Ariane Beland, Stéphanie Vanslambrouck, Lisa Danielczak, Ranjan Roy, Frank Caporuscio, Natalie Fekete and Gad Sabbatier for technical support and Raymond Tran for his help in reviewing the manuscript.

## Author Contributions

**Conceptualization:** Mariève D. Boulanger, Hugo A. Level, Corinne A. Hoesli.

**Data curation:** Mariève D. Boulanger, Hugo A. Level, Mohamed A. Elkhodiry, Omar S. Bashth, Pascale Chevallier, Gaétan Laroche, Corinne A. Hoesli.

**Formal analysis:** Mariève D. Boulanger, Hugo A. Level, Mohamed A. Elkhodiry, Omar S. Bashth, Pascale Chevallier, Gaétan Laroche, Corinne A. Hoesli.

**Funding acquisition:** Gaétan Laroche, Corinne A. Hoesli.

**Investigation:** Mariève D. Boulanger, Hugo A. Level, Mohamed A. Elkhodiry, Omar S. Bashth, Pascale Chevallier.

**Methodology:** Mariève D. Boulanger, Hugo A. Level, Mohamed A. Elkhodiry, Omar S. Bashth, Pascale Chevallier, Gaétan Laroche, Corinne A. Hoesli.

**Project administration:** Gaétan Laroche, Corinne A. Hoesli.

**Resources:** Gaétan Laroche, Corinne A. Hoesli.

**Supervision:** Gaétan Laroche, Corinne A. Hoesli.

**Validation:** Mariève D. Boulanger, Mohamed A. Elkhodiry.

**Visualization:** Mariève D. Boulanger, Hugo A. Level, Mohamed A. Elkhodiry, Omar S. Bashth, Pascale Chevallier.

**Writing – original draft:** Mariève D. Boulanger, Hugo A. Level, Mohamed A. Elkhodiry, Omar S. Bashth, Pascale Chevallier.

**Writing – review & editing:** Mariève D. Boulanger, Hugo A. Level, Mohamed A. Elkhodiry, Omar S. Bashth, Pascale Chevallier, Gaétan Laroche, Corinne A. Hoesli.

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
