## [Decision Letter · Decision Letter 0]

2 Nov 2021

PONE-D-21-26288Bioaffinity-based surface immobilization of antibodies to capture endothelial colony-forming cellsPLOS ONE

Dear Dr. Hoesli,

Thank you for submitting your manuscript to PLOS ONE. After careful consideration, we feel that it has merit but does not fully meet PLOS ONE’s publication criteria as it currently stands. Therefore, we invite you to submit a revised version of the manuscript that addresses the points raised during the review process.

We look forward to receiving your revised manuscript.

Kind regards,

Oksana Lockridge, Ph.D.

Academic Editor

PLOS ONE

Journal Requirements:

a) Did participants provide their written or verbal informed consent to participate in this study?

This study was financially supported by the Canadian Institutes of Health Research (CIHR, MOP 142285; CAH and GL) and the Canadian Foundation for Innovation (CFI, project 35507; CAH). This research was undertaken, in part, thanks to funding from the Canada Research Chairs Program (CAH). This work was supported via travel awards and networking opportunities offered by ThéCell (The Quebec Network for Cell, Tissue and Gene Therapy; MDB, MAE, OSB, CH), the Quebec Center for Advanced Materials (QCAM; MDB, MAE, OSB, GL, CH), PROTEO (The Quebec Network for Research on Protein Function; MDB, MAE, OSB, CH), CMDO (the Cardiometabolic Health, Diabetes, and Obesity Research Network; MDB, MAE, OSB, CH), and the MRM (McGill Regenerative Medicine; MDB, MAE, OSB, CH) network. MDB received a Graduate Excellence Fellowship from the Faculty of Engineering of McGill University. MAE received a scholarship from the Fonds de recherche du Québec - Nature et technologies (Programme de bourses d'excellence pour étudiants étrangers 262907). OSB received a 2017 Hadhramout Establishment for Human Development scholarship. The funders had no role in study design, data collection and analysis, decision to publish, or preparation of the manuscript.

This study was financially supported by the Canadian Institutes of Health Research (CIHR, MOP 142285; CAH and GL) and the Canadian Foundation for Innovation (CFI, project 35507; CAH). This research was undertaken, in part, thanks to funding from the Canada Research Chairs Program (CAH). This work was supported via travel awards and networking opportunities offered by ThéCell (The Quebec Network for Cell, Tissue and Gene Therapy; MDB, MAE, OSB, CH), the Quebec Center for Advanced Materials (QCAM; MDB, MAE, OSB, GL, CH), PROTEO (The Quebec Network for Research on Protein Function; MDB, MAE, OSB, CH), CMDO (the Cardiometabolic Health, Diabetes, and Obesity Research Network; MDB, MAE, OSB, CH), and the MRM (McGill Regenerative Medicine; MDB, MAE, OSB, CH) network. MDB received a Graduate Excellence Fellowship from the Faculty of Engineering of McGill University. MAE received a scholarship from the Fonds de recherche du Québec - Nature et technologies (Programme de bourses d'excellence pour étudiants étrangers 262907). OSB received a 2017 Hadhramout Establishment for Human Development scholarship. The funders had no role in study design, data collection and analysis, decision to publish, or preparation of the manuscript.

Reviewers' comments:

Reviewer's Responses to Questions

**Comments to the Author**

1. Is the manuscript technically sound, and do the data support the conclusions?

Reviewer #1: Yes

Reviewer #2: Yes

2. Has the statistical analysis been performed appropriately and rigorously? 

Reviewer #1: Yes

Reviewer #2: No

3. Have the authors made all data underlying the findings in their manuscript fully available?

Reviewer #1: Yes

Reviewer #2: Yes

4. Is the manuscript presented in an intelligible fashion and written in standard English?

Reviewer #1: Yes

Reviewer #2: Yes

5. Review Comments to the Author

Reviewer #1: The authors investigate the immobilization of EPC capture antibodies on cysteine-tagged protein G grafted polystyrene surface, as well as the ECFCs capture capability of modified surface under dynamic flow conditions. The manuscript was well prepared and some interesting outcomes were obtained. However, more advanced experiments should be included so that the ECFCs capture mechanism of the modified surface can be better clarified. This paper is not recommended for publication until it is properly revised.

1. The orientation of the grafted antibodies was demonstrated playing a major role in ECFCs capture. However, no evidence was provided to verify the orientation of antibodies on the polystyrene surface, which hardly makes the outcomes and corresponding analysis convincing. Therefore, the orientation of antibodies on different modified surface MUST be supplemented.

2. The success of antibodies grafting cannot be confirmed by the content changes of C, N and O element. The high-resolution XPS results should be provided. Also, more advanced characterization should be considered.

3. The ECFCs capture capability of the modified surface should be further studied. The authors can refer to the evaluation methods in the following reference: “Bio-clickable and Mussel Adhesive Peptide Mimics for Engineering Vascular Stent Surfaces”.

Reviewer #2: Overall Opinion

• The paper is written to a high grade and is an eloquently designed study with a detailed methodology and results section.

Overall Comments

• Could you use this technology be adapted to isolate ECFCs from peripheral blood for in vitro expansion as this is often a problem due to the low frequency of ECFCs in PBMCs?

• Is there any evidence to support the idea that ECFCs can proliferate successfully on this novel surface?

• There is a lot of review articles references, the inclusion of a higher number of primary articles is needed.

• It may be worth having a chemical diagram to demonstrate the process + binding (even if it is part of the supplemental material) such as that featured in the following paper, figure 1, which was previously published by your lab. It would diagrammatically demonstrate the binding to the aminated surfaces. https://pubs.rsc.org/en/content/articlepdf/2020/bm/d0bm00650e?page=search Bashth, O.S., Elkhodiry, M.A., Laroche, G. and Hoesli, C.A., 2020. Surface grafting of Fc-binding peptides as a simple platform to immobilize and identify antibodies that selectively capture circulating endothelial progenitor cells. Biomaterials Science, 8(19), pp.5465-5475.

Abstract:

• Authors should mention that their strategy facilitates improved ECFC binding due to improved orientation of capture antibodies.

Introduction:

• Background well described. Each point is explained and referenced before making the next point. However many of the references used are review articles and not primary articles, it would be beneficial to include more primary references for points made if possible

• Line 41: ‘Capturing EPCs, … ‘ should include primary references such as: https://ashpublications.org/blood/article/104/9/2752/19414/Identification-of-a-novel-hierarchy-of-endothelial Ingram, D.A., Mead, L.E., Tanaka, H., Meade, V., Fenoglio, A., Mortell, K., Pollok, K., Ferkowicz, M.J., Gilley, D. and Yoder, M.C., 2004. Identification of a novel hierarchy of endothelial progenitor cells using human peripheral and umbilical cord blood. Blood, 104(9), pp.2752-2760.

• Line 64: What do the authors mean by ‘partial denaturation’ and is this the appropriate term in this case?

• Line 60: ‘Using passive adsorption, …’ may benefit from referring to a primary article along with the already referenced to better support the point being made

• Line 80, 82: Due to the confusion associated with EPCs vs ECFCs that exists in the literature I think it important to keep it consistent after introducing ECFCs, therefore I suggest using ECFCs throughout the manuscript.

Methods:

• Methods section is very well described with detailed description of step by step procedures

• Line 104: minor typo of ‘2N’ where I assume it should be ‘2M’

• Line 106: The authors mention that the antibody coated plates were left to dry for up to one week, is there any idea as to whether there is any degree of antibody denaturation or reduced cell capture with time?

• Line 159: How did the authors confirm that only the FAB part of the chick antibody binds to the protein G?

• Section 2.6: Why was the surface not rinsed with BSA to prevent antibody non-specific binding? There seems to be quite a bit of background fluorescence in the images outside the drop area where there should be no protein G to the point where there is more fluorescence in the background than the lowest dose of protein G.

• There is no evidence that the data was tested for normality.

• For improved reproducibility/comparibility it would be helpful if the authors provided the ECFC donors mean age, N number and genders

Results

• Figure 2)

o Why were only the 1mg/ml and the 3mg/ml included in part B (Orange II concentration) and not 0.5 and 1.5mg/ml treated surfaces?

o There is no explanation as to why 1mg/ml and 3mg/ml were used going forward. Or which one was used for future studies

• Paragraph starting on line 251: Is it possible that S-SMPB is binding more than 1 protein G which is facilitating the higher absorbance?

• Figure 4)

o What S-SMPB concentration was used for this experiment?

• Figure 5)

o Is there more protein G binding outside the drop in the 0.55uM concentration? (Figure Bi) Can the authors suggest a reason for why it would be higher?

o The inclusion of a S-SMPB(-) fluorescence image here as a negative control would also benefit to highlight the improvement in fluorescent signal

o Why continue with CD144 and not CD31? Since the authors decided to use CD144 for the remainder of the study then the images in figure 5B should be of CD144 and not CD31.

Additionally, the authors should include fluorescent images of the other antibodies mentioned in section 3.3 in the supplementary figures section (i.e. anti-CD105, anti-CD144, CD14).

o There are some ‘defects’ in binding in some areas of the droplet resulting in non-homogenous staining, especially in the higher concentrations, and therefore would indicate inhomogeneity in ECFC binding later. Would this affect endothelialisation of a graft with ECFCs if there are regions of non-binding?

o What concentration of protein G was used to generate the graph in part C? and why was this concentration chosen over the others?

• Section 3.4:

o The authors have not explained their reasoning for selecting CD144 over CD31 and CD105 as a capture antibody?

o While I see that using ECFCs at a low shear stress demonstrates an important initial proof of concept of cell capture, I believe an important experiment that would need consideration is passing PBMCs over the treated surface to determine ECFC capture from a mixed cell population and at rate at a higher and more physiological shear stress for longer periods of time. i.e. to better mimic in vivo conditions and to demonstrate proof of concept in a more relevant in vitro model. This would also better facilitate moving into an animal model. Additionally, due to the potential inflammatory effect of the protein G this experiment could also double up to demonstrate immune cell activation.

• Figure 6)

o Interesting that the ECFCs indicated by the green triangle seem to have lower binding than the others in all conditions. It might be worthwhile measuring CD144 expression on the ECFCs of that donor relative to the others.

• Figure S1

o Was there any endothelial specific marker (not CD144) used to identify any ECFC binding to the surface? It will be problematic if only immune cells are binding to the surface, this may prevent any ECFCs binding to the surface.

Discussion

• While the discussion discusses the feasibility of the strategy used in this paper and compares it to a past paper providing a balanced discussion, it does not compare it to other modification strategies used by other papers/groups. An improved literature comparison would benefit this section.

• Line 324: Due to the confusion that remains in the field it may be best to consistently use the term ‘ECFC’ instead of ‘EPC’ as EPC can be thought of as a cell with immune cell properties.

• Line 355: Change EPC to ECFC

Conclusion

• Line 360: Change ‘endothelial progenitor cell’ to ‘ECFC’

• Line 369: Change ‘EPC’ to ‘ECFC’

6. PLOS authors have the option to publish the peer review history of their article (what does this mean?). If published, this will include your full peer review and any attached files.

Reviewer #1: No

Reviewer #2: **Yes: **Caomhán J. Lyons

---

## [Author Response · Author response to Decision Letter 0]

9 May 2022

Please refer to the attached document which contains image that are not pasted below.

----------------

Dear PLOS ONE editor and reviewers,

We thank you for the time you invested reviewing our manuscript titled “Bioaffinity-based surface-immobilization of antibodies to capture endothelial colony-forming cells”. We appreciate your patience in receiving this revision caused by the aftershocks of the pandemic, graduation of all students (Elkhodiry, Bashth & Boulanger) that previously performed the work, and stock shortage (over 4 months lead time) in the PureCoat® Amine plates we used for previous experiments. 

We appreciated insightful comments on our claims related to oriented immobilization of antibodies and interpretation of XPS results which led to a significant revision of the content of this manuscript. The revised manuscript has been significantly modified and improved through the main changes listed below:

1. Addition of high-resolution XPS data (new Figure 2)

2. Successful Protein G immobilization on another aminated polystyrene substrate which could address future supply issues in PureCoat® Amine plates, which as been a long-standing issue in our experience (new Figure A2 in Appendix S1)

3. More cautious and rigorous assessment of our claim that antibodies are oriented on surfaces to focus rather on affinity-mediated interactions vs other immobilization methods. We also provide proof-of-concept AFM studies indicating that at least some antibodies are indeed oriented on surfaces (new Figure A4 in Appendix S1).

4. Modification of most figures, including Figure 1 (improved graphical representation), the Striking Image (as requested by Reviewer 2) and panel B in Figure 4 (anti-CD144 added, S-SMPB(-) controls added in new Figure A3 in the appendix).

Specific response to comments by Reviewer 1

Overall: “The authors investigate the immobilization of EPC capture antibodies on cysteine-tagged protein G grafted polystyrene surface, as well as the ECFCs capture capability of modified surface under dynamic flow conditions. The manuscript was well prepared and some interesting outcomes were obtained. However, more advanced experiments should be included so that the ECFCs capture mechanism of the modified surface can be better clarified. This paper is not recommended for publication until it is properly revised.”

Response: We thank Reviewer 1 for noting that our manuscript was carefully prepared. We believe that our additional studies (AFM, high-resolution XPS data, demonstration that protein G can be immobilized on other commercial aminated polystyrene substrates) help clarify how antibodies are immobilized and interact with ECFC surface antigens.

Reviewer 1, comment 1: “The orientation of the grafted antibodies was demonstrated playing a major role in ECFCs capture. However, no evidence was provided to verify the orientation of antibodies on the polystyrene surface, which hardly makes the outcomes and corresponding analysis convincing. Therefore, the orientation of antibodies on different modified surface MUST be supplemented.”

Response: We thank the reviewers for raising this question, which is at the core of the hypothesis tested in this work. Determining which regions of antibodies interact with different domains of protein G is not trivial and has been the object of numerous previous studies since the 1980s (1-9). Streptococcal protein G has a high promiscuity towards mammalian IgG Fc fragment binding. The Fc-binding domains of protein G can also interact with Fab regions of IgGs of certain species or antibody subtypes, typically with lower affinity than the strong Fc interactions.

The Fc-mediated interactions reportedly allow oriented antibody immbolization on surfaces (10-12). A study by Young Min Bae et al. (10) demonstrated that surface grafting of antibodies through the Fc-binding regions of protein G leads to a higher immobilized antibody concentration when compared to a control surface (i.e. without protein G), which corroborates our results. This study also proposed an immobilization profile of antibodies based on surface plasmon resonance (SPR) and atomic force microscopy (AFM) experiments. 

We performed a proof-of-concept AFM study of our different surfaces. The methods and results of this experiment are shown below and were added to the Supplementary Information. We were able to qualitatively observe an increase in the density of features on the surface with protein G immobilization, compared to antibody adsorption (omission of the S-SMPB and protein G polypeptide steps). AFM depth histograms were compared for the different conditions, showing a noticeable shift toward higher relative depth with antibody immobilization on protein G.

We recognize that our experiments, despite the addition of the AFM data, do not directly address antibody orientation on surfaces. To clarify this question, we have added further details on the recombinant protein G polypeptide and its known interactions with different IgG domains.

Added text in the abstract:

A cysteine-tagged truncated protein G polypeptide containing three Fc-binding domains was conjugated onto aminated polystyrene substrates via a bi-functional linking arm, followed by antibody immobilization.

Added text in materials and methods:

Next, the cysteine-tagged protein G sequence was attached to the linking arm by adding 150 µL/cm2 of a 5.5 µM recombinant cysteine-tagged protein G polypeptide (henceforth termed “protein G polypeptide” - a recombinant non-glycosylated polypeptide chain produced in E. coli containing an N-terminal Cys followed by amino acids 298-497 of the streptococcal protein G sequence, #PRO-1328, Prospec-Tany Technogene Ltd) suspension in PBS for 1 h.

Added discussion:

The protein G polypeptide used in this study comprises amino acids 298-497 of the full streptococcal protein G sequence, which includes the three Fc-binding domains of this protein. Streptococcal protein G interacts with immunoglobulin G antibodies of most mammalian species (38, 39). This interaction can be mediated both through Fc and Fab regions (40), although the association constant is 1 order of magnitude higher for Fc fragments (41) with the exception of mouse IgG1 where Fab plays a more significant role (42, 43). A polypeptide fragment containing the sequence used in the current study was previously shown to interact specifically with the Fc and not the Fab (40, 44) region of human IgG antibodies. All antibodies in the current work were of IgG1 subclass except anti-CD144, of IgG2a subclass. It is possible that the difference in subclass led to different antibody interactions with adsorbed vs conjugated protein G observed in Figure 5. Theoretically, the higher ratio between Fc:Fab protein G binding affinity expected for IgG2 antibodies would lead to improved orientation on surfaces. However, even for antibodies where Fab interactions with protein G are significant, the hypervariable region is expected to remain available for antigen recognition (45).

Added supplementary information:

3. Atomic Force Microscopy (AFM) imaging of modified polystyrene

A commercial aminated petri dish was cut into 1 cm*1 cm coupons and then functionalized according to the protocol previously described. Briefly, the coupons were covered with 3mg/mL Sulfo-SMPB for 2 hours, rinsed with PBS and then reacted with 5.5 uM Cys-Protein G. Primary antibodies (anti-CD309 mouse IgG1) were then immobilized on the surface at 5 µg/mL for 1 hour. The surfaces were finally rinsed twice with PBS and RO water before air drying and AFM imaging. An unmodified aminated polystyrene coupon was analyzed as a control. The images were acquired both in ambient conditions using a NanoscopeV-Dimension ICON atomic force microscope (Bruker, Santa Barbara, CA, USA). All Imaging was done in the PeakForce Tapping mode (PeakForce QNM®) using SCANASYST-Fluid aluminum-coated Silicon Nitride probes with tip radius ranging between 2-10 nm and a nominal spring constant of 0.4 N/m. The scanning rate was 1 Hz and for each condition, two images were taken with two different resolutions, 5 µm x 5 µm and 1 µm x 1 µm.

Image treatment and analysis was performed with the NanoScope software (Bruker). The arithmetic roughness average was computed for the 1 um2 images to avoid the surface defects observed at larger scales. The depth distribution of a 500 nm x 500 nm area was also obtained for each coupon.

Figure A4. AFM imaging of modified polystyrene surfaces. (A) 5 um2 images of an unmodified polystyrene coupon (aminated PureCoatTM amine), antibody adsorption (surface functionalization omitting S-SMPB) and anti-CD309 antibody immobilization on protein G (complete functionalization scheme). (B) 1 um2 images of the same conditions. The white squares represent the 500 nm2 areas that were used to obtain the surface depth histograms. (C) Surface roughness (arithmetic average, Ra) obtained from the 1 um2 images to avoid surface defects visible at higher scales. (D) Depth distribution profiles obtained from the raw height data of 500 nm2 areas on each image. The vertical red line at 15 nm marks the typical length of an antibody.

Reviewer 1, comment 2: “The success of antibodies grafting cannot be confirmed by the content changes of C, N and O element. The high-resolution XPS results should be provided. Also, more advanced characterization should be considered.”

Response: We thank the reviewers for this comment. Indeed, in the previous version of the manuscript, we didn’t include the XPS analyses for protein G and antibody modification. Based on this comment, we added the survey data associated with these surface modification steps, as well as high resolution spectra of C1s. 

Modifications to materials and methods:

The chemical composition of the aminated surfaces, before and after functionalization was investigated by XPS using a PHI 5600-ci spectrometer (Physical Electronics, Eden Prairie, MN). The main XPS chamber was maintained at a base pressure of < 8×10-9 Torr. A standard aluminum X-ray (Al Kα = 1486.6 eV) source was used at 300 W to record survey spectra with charge neutralization, while C1s high resolution spectra were recorded with a standard magnesium X-ray source without neutralization. The detection angle was set at 45º with respect to the normal of the surface and the analyzed area was 0.5 mm2. The spectrometer work function was adjusted to 285.0 eV for the main C (1s) peak. Curve fitting of high-resolution peaks were determined by means of the least squares minimization procedure employing Gaussian-Lorentzian functions and a Shirley-type background.

Modifications to the results section:

Subsequent protein G polypeptide grafting led to an increase in the O and N content related to the presence of these atoms in amino acid side chains and C terminus. The penetration depth of XPS analysis is ~5 nm which is inferior to the size of antibodies and on the same range as the size of a ~22 kDa polypeptide (~2 nm expected size) such as the protein G sequence used here. Atomic ratios therefore provide a useful metric to determine whether protein G polypeptide and antibody immobilization steps were successful. The O/C and N/C ratios decreased when anti-CD144 antibodies were immobilized on the surface. This observation suggests that anti-CD144 antibodies have higher C-rich amino acid content than the protein G polypeptide. This observation which was corroborated by C1s high-resolution spectra (Figure 2): the peak at 285.0 eV, associated to C-C/C-H bonds, reached 76% for CD144 surface whereas it was 65% on Cys-Protein G.

Table 1. Surface atomic composition assessed by XPS survey analyses*. 

Reaction step after which XPS analysis was conducted Atomic percentage Atomic ratios

 %C %O %N N/C O/C

Initial pureCoatTM amine polystyrene surfaces 63 ± 2 17.2 ± 0.9 18.8 ± 0.6 0.30 ± 0.02 0.26 ± 0.02

After S-SMPB 86 ± 1 7.6 ± 0.9 5.0 ± 0.5 0.057 ± 0.006

 0.09 ± 0.01

After Cys-protein G 83.3 ± 0.9 9.7 ± 0.7 7.0 ± 0.4 0.084 ± 0.006 0.116 ± 0.009

After antibody immobilization 85.8 ± 0.9 8.5 ± 0.9 4.7 ± 1.1 0.05 ± 0.01 0.09 ± 0.01

* Error estimates represent the standard deviation of areas analyzed on one sample to assess the grafting homogeneity.

Figure 1. High Resolution C1s XPS spectra. (A) Surfaces analyzed after the protein G polypeptide graftin step. (B) Surfaces after the antibody immobilization step using anti-CD144 antibodies.

Reviewer 1 Comment 3: “The ECFCs capture capability of the modified surface should be further studied. The authors can refer to the evaluation methods in the following reference: “Bio-clickable and Mussel Adhesive Peptide Mimics for Engineering Vascular Stent Surfaces”.”

Response: We view this study as a first step towards the development of a variety of cell capture applications using protein G-mediated antibody immobilization. We agree that more in-depth studies of ECFC behaviour on surfaces would be valuable for capture stent or other vascular biomaterials applications. However, our preliminary data suggests that combination of bioaffinity-immobilized antibodies with integrin-binding peptides will be needed to create a suitable environment for ECFC firm adhesion, cell spreading and proliferation (13). We have reported these findings in a patent application and plan to publish this work in the upcoming year, building on the findings published in the current work.

We propose avenues for future development of the reported protein G-based strategy in the following added text in the discussion:

Given the versatility of the proposed bioaffinity-based antibody immobilization strategy as demonstrated through immobilization of IgGs targeting different cell antigens, it would be interesting to study selective capture of different immune cell subsets from peripheral blood. The proliferation of ECFC post-capture should also be assessed. Combination of Fc-binding peptides or polypeptides with integrin-binding peptides shows significant promise in this regard (49). 

 

Specific response to comments by Reviewer 2

Overall opinion: “The paper is written to a high grade and is an eloquently designed study with a detailed methodology and results section.”

We thank the reviewer for their appreciation of our efforts.

Reviewer 2, Comment 1: “Could you use this technology be adapted to isolate ECFCs from peripheral blood for in vitro expansion as this is often a problem due to the low frequency of ECFCs in PBMCs?”

Response: We believe that the bioaffinity-based approaches we presented herein as well as in previous work (14) could be used as a first step towards improving ECFC isolation from blood. Given the sparce nature of peripheral blood ECFCs, we believe that combination with integrin-binding peptides which drive their clonal expansion as we have previously demonstrated (15) could improve primary ECFC isolation yields. We recently filed an international patent application (13) which describes this bi-functional approach. We have not yet tested the idea of using these surfaces for primary colony isolation, but this would be of significant value.

These considerations were added to the “Discussion” section of the manuscript:

Given the versatility of the proposed bioaffinity-based antibody immobilization strategy as demonstrated through immobilization of IgGs targeting different cell antigens, it would be interesting to study selective capture of different immune cell subsets from peripheral blood. The proliferation of ECFC post-capture should also be assessed. Combination of Fc-binding peptides or polypeptides with integrin-binding peptides shows significant promise in this regard (49). This bi-functionnal coating could also be applied on various substrates to efficiently isolate ECFC from peripheral blood in vitro, paving the way to the development of novel cell-culture materials.

Reviewer 2, Comment 2: “Is there any evidence to support the idea that ECFCs can proliferate successfully on this novel surface?”

Response: To our knowledge, there is no evidence of such proliferation activity of ECFC on such substrates – as shown in Figures 14 & 15 of our recent patent application (13). We also believe that there is no obvious reason for immobilized capture antibodies or the Fc-binding protein G fragment to trigger biological effects in ECFC that would enhance their proliferation. In fact, the idea of a bi-functional surface described in our patent application relies on the addition of another molecule (e.g. the RGD peptide), whose role would be to enhance cellular proliferation on the surface. We have drafted a separate manuscript showing most of the results described in the patent application on bi-functional surface modifications which we plan to submit very shortly.

Reviewer 2, Comment 2: “There is a lot of review articles references, the inclusion of a higher number of primary articles is needed.”

Response: Thank you for this comment. We have added the following primary references:

7. Ingram DA, Mead LE, Tanaka H, Meade V, Fenoglio A, Mortell K, et al. Identification of a novel hierarchy of endothelial progenitor cells using human peripheral and umbilical cord blood. Blood. 2004;104(9):2752-60.

16. Butler JE, Ni L, Brown WR, Joshi KS, Chang J, Rosenberg B, et al. The immunochemistry of sandwich ELISAs--VI. Greater than 90% of monoclonal and 75% of polyclonal anti-fluorescyl capture antibodies (CAbs) are denatured by passive adsorption. Mol Immunol. 1993;30(13):1165-75.

42. Bjorck L, Kronvall G. Purification and some properties of streptococcal protein G, a novel IgG-binding reagent. J Immunol. 1984;133(2):969-74.

43. Boyle MD. CHAPTER 1 - Introduction to bacterial immunoglobulin-binding proteins,. In: Boyle MD, editor. Bacterial Immunoglobulin-Binding Proteins: Academic Press; 1990. p. 1-21.

44. Erntell M, Myhre EB, Sjobring U, Bjorck L. Streptococcal protein G has affinity for both Fab- and Fc-fragments of human IgG. Mol Immunol. 1988;25(2):121-6.

45. Stone GC, Sjobring U, Bjorck L, Sjoquist J, Barber CV, Nardella FA. The Fc binding site for streptococcal protein G is in the C gamma 2-C gamma 3 interface region of IgG and is related to the sites that bind staphylococcal protein A and human rheumatoid factors. J Immunol. 1989;143(2):565-70.

46. Derrick JP, Wigley DB. The third IgG-binding domain from streptococcal protein G. An analysis by X-ray crystallography of the structure alone and in a complex with Fab. J Mol Biol. 1994;243(5):906-18.

47. Kato K, Lian LY, Barsukov IL, Derrick JP, Kim H, Tanaka R, et al. Model for the complex between protein G and an antibody Fc fragment in solution. Structure. 1995;3(1):79-85.

48. Derrick JP, Wigley DB. Crystal structure of a streptococcal protein G domain bound to an Fab fragment. Nature. 1992;359(6397):752-4.

49. Erntell M, Myhre EB, Kronvall G. Alternative non-immune F(ab')2-mediated immunoglobulin binding to group C and G streptococci. Scand J Immunol. 1983;17(3):201-9.

Reviewer 2, Comment 2: “It may be worth having a chemical diagram to demonstrate the process + binding (even if it is part of the supplemental material) such as that featured in the following paper, figure 1, which was previously published by your lab. It would diagrammatically demonstrate the binding to the aminated surfaces. https://pubs.rsc.org/en/content/articlepdf/2020/bm/d0bm00650e?page=search

Bashth, O.S., Elkhodiry, M.A., Laroche, G. and Hoesli, C.A., 2020. Surface grafting of Fc-binding peptides as a simple platform to immobilize and identify antibodies that selectively capture circulating endothelial progenitor cells. Biomaterials Science, 8(19), pp.5465-5475.”

Response: We have updated Figure 1 to clarify the surface modification and detection steps. We have also updated the “striking figure” which shows both the surface modification process & cell binding as suggested above. The new figures are shown below.

Figure 1. Schematic representation of the antibody immobilization process followed by a fluorescence-based antibody detection step.

Striking Figure (graphical abstract).

Reviewer 2, Comment on Abstract: “Authors should mention that their strategy facilitates improved ECFC binding due to improved orientation of capture antibodies.”

Response: We thank the reviewers for this comment. Although we expect the Fc-binding interaction to be privileged and therefore to promote a better antibody orientation (as it has been shown elsewhere), the Fc-binding domains of protein G can also interact with the more constant portions of Fab depending on species and antibody subclass as detailed in our response to Reviewer 1, Comment 1. Albeit our AFM data suggests higher fraction of oriented antibodies vs direct adsorption, what is more clear is that ECFC capture was significantly enhanced with our technology. We also achieved a higher antibody coverage than with adsorption (as shown in the revised Figure 4 and in the AFM image in Supplementary Information). We believe that multiple factor in our antibody immobilization strategy can enhance antibody binding and cell capture (concentration, stability, distance created by the protein G…), orientation being only one of those. We were therefore quite cautious to avoid overstating orientation in the abstract and revised manuscript.

Reviewer 2, Comment on Introduction: “Background well described. Each point is explained and referenced before making the next point. However many of the references used are review articles and not primary articles, it would be beneficial to include more primary references for points made if possible.”

Response to Reviewer 2, Comment on Introduction: Thank you for this comment, we added more primary references as described in the response to Comment 2.

Reviewer 2, Comment on Line 41:” ‘Capturing EPCs, … ‘ should include primary references such as:

https://ashpublications.org/blood/article/104/9/2752/19414/Identification-of-a-novel-hierarchy-ofendothelial

Ingram, D.A., Mead, L.E., Tanaka, H., Meade, V., Fenoglio, A., Mortell, K., Pollok, K.,

Ferkowicz, M.J., Gilley, D. and Yoder, M.C., 2004. Identification of a novel hierarchy of endothelial progenitor cells using human peripheral and umbilical cord blood. Blood, 104(9), pp.2752-2760.”

Response: This reference was added as an obvious omission.

Reviewer 2, Comment on Line 64: “What do the authors mean by ‘partial denaturation’ and is this the appropriate term in this case?”

Response: What we meant in this statement was that proteins in contact with surfaces can unfold at least partially due to the so-called hydrophobic effect and interactions with surface functional groups (16). When using crosslinking reagents for direct covalent antibody immobilization (via amine or hydroxyl functional groups), multiple chemical modifications can occur on different amino acids in the antibody sequence, some of which can be near or part of the antigen recognition site, therefore reducing its binding capacity by changing its conformation/structure.

We have rephrased our statement as follows:

The abundance of these functional groups in an antibody results in random antibody orientations on the surface and could lead to changes in conformation affecting its antigen binding efficacy (18, 19). Unwanted reactions between amino acids in hypervariable region of the antibody and the cross-linking reagents used for covalent immobilization could also directly affect the antigen-binding capacity of the antibody.

Reviewer 2, Comment on Line 60: “ ‘Using passive adsorption, …’ may benefit from referring to a primary article along with the already referenced to better support the point being made.”

Response: We have added the following reference:

16. Butler JE, Ni L, Brown WR, Joshi KS, Chang J, Rosenberg B, et al. The immunochemistry of sandwich ELISAs--VI. Greater than 90% of monoclonal and 75% of polyclonal anti-fluorescyl capture antibodies (CAbs) are denatured by passive adsorption. Mol Immunol. 1993;30(13):1165-75.

Reviewer 2, Comment on Lines 80, 82: “Due to the confusion associated with EPCs vs ECFCs that exists in the literature I think it important to keep it consistent after introducing ECFCs, therefore I suggest using ECFCs throughout the manuscript.”

Response: We are aware of the confusion surrounding EPC terminology. However, the literature surrounding stents that use immobilized antibodies to capture cells of endothelial phenotype broadly uses the term “EPC capture stents”. The only clinically-approved endothelial capture stent, commercialized by Orbus-Neich, utilizes this terminology. We believe that the term is appropriate in this context since any given antibody will unlikely be specific only towards ECFCs. The antibody immobilization strategy we use in this manuscript could, in fact, also be used to capture myeloid angiogenic cells which can also express markers such as CD31 or CD144, at least in culture.

Therefore, we have retained the use of “EPC” when referring to capture technologies, but have been very careful in using “ECFCs” when referring to bona fide progenitors, as well as the cell model used in the current study. We also added a reference to the consensus statement on nomenclature in the introduction.

Changes:

Oriented surface immobilization of antibodies via covalent grafting of cysteine tagged protein G remains untested for in vivo cell capture applications, particularly ECFCs.

in order to immobilize antibodies which recognize endothelial surface markers such as CD31 and CD144 (replaced “EPC capture antibodies” in this sentence)

Reviewer 2 Comment on Methods: “Methods section is very well described with detailed description of step by step procedures.” 

Response: Thank you for your appreciation.

Reviewer 2 Comment on Line 104: “minor typo of ‘2N’ where I assume it should be ‘2M’ “

Response: “2N” refers to the normality of the solution used (which, in the case of NaOH, would also correspond to the molar concentration). This product was in fact purchased as a “2N sodium hydroxide solution”.

Reviewer 2 Comment on Line 106: “The authors mention that the antibody coated plates were left to dry for up to one week, is there any idea as to whether there is any degree of antibody denaturation or reduced cell capture with time?”

Response: We agree that further investigation of the effect of drying and various storage conditions on changes in antibody conformation, stability, antigen and cell binding capacity would be of high interest. We think that drying the surfaces is the most practical approach for most applications including cell capture stents or culture surfaces. It will be important to study these aspects in future work – we thank the reviewer for this suggestion.

Reviewer 2, Comment on line 159: “How did the authors confirm that only the FAB part of the chick antibody binds to the protein G?”

Response: The affinity of protein G toward the Fc fragment of chicken IgY is considered to be negligible. It has also been reported in the industry that avian egg yolk antibodies cannot be isolated via the conventional protein G immobilization technique (widely used for IgG for instance).

The following references were added to this statement in the manuscript:

30. Lee W, Atif AS, Tan SC, Leow CH. Insights into the chicken IgY with emphasis on the generation and applications of chicken recombinant monoclonal antibodies. Journal of Immunological Methods. 2017;447:71-85.

31. Schade R, Staak C, Hendriksen C, Erhard M, Hugl H, Koch G, et al. The production of avian (egg yolk) antibodies: IgY - The report and recommendations of ECVAM Workshop 21. Atla-Altern Lab Anim. 1996;24(6):925-34.

Reviewer 2 Comment on Section 2.6 (now 2.4): “Why was the surface not rinsed with BSA to prevent antibody non-specific binding? There seems to be quite a bit of background fluorescence in the images outside the drop area where there should be no protein G to the point where there is more fluorescence in the background than the lowest dose of protein G.”

Response: We compared different blocking methods including BSA, Dako Protein Block and no blocking agent added and did not observe any significant differences in signal:noise ratio with these agents on the aminated substrates.

The region where protein G is spotted indeed leads to lower adsorption of secondary antibodies as can be seen in Appendix S1 Figure A3. One likely explanation is that this region was in a sense “blocked” by the protein G polypeptide without providing sufficient surface density for efficient Fc-mediated binding.

Reviewer 2 Comment on Statistics: “There is no evidence that the data was tested for normality.”

Response: The data was tested for normality using the distribution/continuous fit/normal distribution/goodness-of-fit platform in JMP(R) 15.1.0. For all datasets, the Shapiro-Wilk test (as well as the Anderson-Darling test) did not reveal any significant departure from normality. We have added the following text to the Statistical Methods section:

“The Shapiro-Wilk normality test was applied prior to performing parametric tests.”

Reviewer 2 Comment on ECFC donor source: “For improved reproducibility/comparibility it would be helpful if the authors provided the ECFC donors mean age, N number and genders”

Response: Thank you for this suggestion. We believe that the reviewer was referring to biological sex and not gender. The sex, mean age and number of donors have been included in the manuscript (the age of each donor is not available as it could allow identification of individual donors known to our colleagues, and because some donors declined to provide age):

(N=4, 2 females, 2 males, mean age : 25.5)

Reviewer 2 Comment on Results, Figure 2 (now Figure A1 in Appendix S1): “Why were only the 1 mg/ml and the 3mg/ml included in part B (Orange II concentration) and not 0.5 and 1.5mg/ml treated surfaces? There is no explanation as to why 1 mg/ml and 3 mg/ml were used going forward. Or which one was used for future studies.”

Response: Thank you for this relevant question. We initially tested several concentrations of S-SMPB to potentially reduce reagent use compared to the conventional amount we have used in the past (3 mg/mL). In the end, we decided to retain this higher concentration ensure a maximal coverage on the surface. 

However, we understand that it may create some confusion and we decided to remove data with S-SMPB concentrations other than 3 mg/mL from the main manuscript – moving this information to Supplementary Information. We believe this could still be useful for users who may want to reduce cost in these studies.

In that context, we replaced the XPS histogram by a table (Table 1) to improve readability and provide error estimates. High resolution XPS spectra were added, as mentioned in our response to Reviewer 1, Comment 2.

We added the following text to the results section:

“All experiments were conducted at 3 mg/mL S-SMPB in this study, but this concentration could potentially be reduced based on the Orange II and water contact angle results.”

Reviewer 2, Comment on paragraph starting on line 251: “Is it possible that S-SMPB is binding more than 1 protein G which is facilitating the higher absorbance?”

Response: It is not possible for a protein-bound Sulfo-SMPB molecule to further react with another thiol group. Once the maleimide functional group reacts with a free thiol (via a Michael addition), it is modified in a way that prevents a second addition (saturated thiosuccinimide). We clarified this in the dotted “inset” in the new version of Figure 1 pasted below.

Reviewer 2, Comment on Figure 4: “What S-SMPB concentration was used for this experiment?”

Response: Please see the answer to the question on Figure 2. A solution of 3 mg/mL of S-SMPB was used for all experiments shown in the main manuscript.

Reviewer 2, Comment on 1 Figure 5 (now Figure A3 in Supplementary Information): “Is there more protein G binding outside the drop in the 0.55uM concentration? (Figure Bi) Can the authors suggest a reason for why it would be higher?”

Response: Thank you for this insightful question. We were also initially puzzled by this phenomenon but in retrospect this is not surprising assuming that the addition of protein G polypeptide changes the surface free energy, and thus changes protein adsorption behaviour.

The experiment design described for this figure implies that no protein G is present outside the drop (which is a 0.55 µM Cys-Protein G drop). The signal you are mentioning therefore comes from adsorption of the primary or of the fluorescent secondary (or both) antibodies onto the substrate. At low protein G polypeptide surface grafting, the blocking effect of the covalently attached protein G may have more negative effects on adsorption than its positive effects on Fc-mediated primary antibody immobilization. At higher protein G polypeptide concentration, the adsorption effects outside the “spotted” area remain unchanged, but the bioaffinity antibody immobilization effect increases.

In short, at low concentrations of protein G polypeptide, we believe that the spotted protein G blocks protein adsorption which is not the case outside the spot area.

Reviewer 2, Comment 2 on Figure 5 (now Figure A3 in Supplementary Information): “The inclusion of a S-SMPB(-) fluorescence image here as a negative control would also benefit to highlight the improvement in fluorescent signal.”

Response: We modified the figure to include a S-SMPB(-) control, as shown below.

Figure A3. Effect of protein G polypeptide concentration in solution on anti-CD31 immobilization via conjugated vs adsorbed protein G. PureCoatTM aminated surfaces were functionalized with protein G polypeptides either via covalent conjugation (S-SMPB(+)) or adsorption (S-SMPB(-)). The protein G polypeptides were applied at different concentrations in solution as indicated in white text on each panel. Mouse IgG1 anti-CD31 antibodies were then added and detected through fluorophore-labelled anti-mouse secondary antibodies. Each condition was applied at least 3 times. Representative spots are shown. Dotted yellow lines represent spot contours.

Reviewer 2, Comment 2 on Figure 5 (now Figure 4 & Figure A3): “Why continue with CD144 and not CD31? Since the authors decided to use CD144 for the remainder of the study then the images in figure 5B should be of CD144 and not CD31. Additionally, the authors should include fluorescent images of the other antibodies mentioned in section 3.3 in the supplementary figures section (i.e. anti-CD105, anti-CD144, CD14).”

Response: We now include both anti-CD31 and anti-CD144 in the main manuscript Figure 4. CD31 is expressed on endothelial cells but also at low levels on platelets, granulocytes, macrophages, dendritic cells and other lymphocytes. CD144, which labels endothelial cell cadherins, is expected to be more selective for cells of endothelial phenotype.

We added the images for anti-CD144 to Figure 4 but decided to keep the CD31 images as well as a point of comparison between the behaviour of IgG1 (e.g. the anti-CD31 used here) and IgG2a (e.g. the anti-CD144 antibody used here) antibodies, given the higher reported Fab binding attributes of IgG1 vs IgG2a. We think adding images for anti-CD105 and anti-CD14 in this figure would be redundant with panel C of the figure (very similar behaviour to anti-CD31). However, if a second revision is needed, these images could be added in Supplementary Information.

In addition to the updated Panel B, we also updated Panel A to reflect the new graphics shown in Figure 1. The modified Figure 4 is shown below:

Reviewer 2, Comment 3 on Figure 5 (now Figure 4 & Figure A3): There are some ‘defects’ in binding in some areas of the droplet resulting in non-homogenous staining, especially in the higher concentrations, and therefore would indicate inhomogeneity in ECFC binding later. Would this affect endothelialisation of a graft with ECFCs if there are regions of non-binding?

Response: We have observed heterogeneous fluorescence when immobilizing not only antibodies via Protein G, but also when grafting TRITC-labeled RGD peptides directly onto the Corning PureCoat(R) Amine substrates (15). We also observed heterogeneity when working with other commercial substrates such as silane coated glass slides (Electron Microscopy Sciences) (17). Much more homogeneous results were obtained when grafting fluorophore-labeled peptides on in-house plasma treated aminated PTFE substrates (18, 19). 

Together, our results point towards non-homogeneous distribution of primary amines on many commercially-available substrates, including the PureCoat(R) Amine brand from Corning.

Reviewer 2, Comment 3 on Figure 5 (now Figure 4 & Figure A3): “What concentration of protein G was used to generate the graph in part C? and why was this concentration chosen over the others?”

Response: Analysis of the Protein G grafting as shown that the concentration of protein G is higher on the surface when a solution with a concentration of at least 5.5 μM was used as shown in Figure 3. No significant differences in primary antibody surface concentration as detected by fluorophore-labelled secondaries was noted between 5.5 μM and 55 μM protein G polypeptide concentration. In order to reduce the use of unnecessary protein G, we performed further experiments with 5.5 μM protein G polypeptide concentration.

We added the following text to Section 3.3:

“A concentration 5.5 μM of protein G was selected for further studies given the higher background signal observed when applying 55 µM protein G polypeptide on S-SMPB(-) surfaces (Figure A3 in Appendix S1).”

Reviewer 2, Comment 1 on Section 3.4: “The authors have not explained their reasoning for selecting CD144 over CD31 and CD105 as a capture antibody?”

Response: To target the specific capture of ECFCs on the modified surface, we needed endothelial cell markers that could target specifically this type of cells. ECFCs are known to express endothelial cell markers, such as CD144 (VE-cadherin), and CD31 (PECAM). CD31 is highly expressed by ECFCs, however, it is also expressed by PBMCs and other potentially other blood circulating cells. We have decided to use CD144 because this marker is more specific to ECFCs even if this marker is less expressed than CD31. 

Reviewer 2, Comment 2 on Section 3.4: “While we see that using ECFCs at a low shear stress demonstrates an important initial proof of concept of cell capture, I believe an important experiment that would need consideration is passing PBMCs over the treated surface to determine ECFC capture from a mixed cell population and at rate at a higher and more physiological shear stress for longer periods of time. i.e. to better mimic in vivo conditions and to demonstrate proof of concept in a more relevant in vitro model. This would also better facilitate moving into an animal model. Additionally, due to the potential inflammatory effect of the protein G this experiment could also double up to demonstrate immune cell activation.”

Response: We agree that higher wall shear stress would be more physiologically-relevant. Low shear stress studies have been broadly used for in vitro ECFC capture studies to allow quantification of statistically significant cell numbers in time frames that do not lead to ECFC death in circulation loops. We have developed an ex vivo flow loop setup which may be useful in future experiments to improve translation.

In addition, we think that the cell capture conditions applied could be relevant to in vitro cell applications such as diagnostics where the wall shear stress levels applied here would remain relevant.

We do envision studying ECFC adhesion and alignment dynamics under high levels of shear stress on our bi-functional antibody/cell adhesion peptide-modified surfaces in future work, as we previously did with mature endothelial cells (20).

Reviewer 2, Comment 1 on Figure 6: Interesting that the ECFCs indicated by the green triangle seem to have lower binding than the others in all conditions. It might be worthwhile measuring CD144 expression on the ECFCs of that donor relative to the others.

Response: Thank you for this interesting suggestion. We agree that this is a striking and consistent trend, especially since the 4 surfaces were studied in “blocks” (i.e. this is not an experiment-to-experiment difference). From our experience with peripheral blood-derived ECFCs, donor to donor variability clearly plays a major role in the physiology of the cells. However, we have not observed significant donor-to-donor variability in CD31 and CD144 expression percentage. Seeing the results presented in figure 6, it would indeed be appropriate to examine mean fluorescence intensity levels of CD144 in more in-depth flow cytometry studies for these specific donors. However, the overall significant improvement of ECFC adhesion observed in our work didn’t support the need to perform such an experiment in the context of that specific study. We will certainly consider acquiring and reporting MFI values in the future if we observe similar trends.

Reviewer 2, Comment 1 on Figure S1 (now Figure A5 in Appendix S1): “Was there any endothelial specific marker (not CD144) used to identify any ECFC binding to the surface? It will be problematic if only immune cells are binding to the surface, this may prevent any ECFCs binding to the surface.”

Response: Experiment demonstrated in Figure S.1. was performed as a control to confirm the capture specificity of a cellular type. For this experiment a population of PMBCs, rich in CD14+ cells (>45%) and with a low CD144+ cell population (˂0.1%) was flown over a surface with immobilized anti-CD144 and anti-CD14. This experiment demonstrated that PBMCs were captured in a higher number on the surface with immobilized anti-CD14 compared to anti-CD144. As you noted, it would be problematic if only immune cells bind to the surface. First, modified surfaces should immobilize only antibodies that are specific to ECFCs and further work should include an extended cell characterization.

We do not think that the number of ECFCs in this study would be sufficient for detection on surfaces. Future studies could apply mixtures of enriched ECFCs and PBMCs to achieve sufficient detectable ECFC numbers. Pre-labelling the ECFCs could be a promising avenue vs detection through another secondary antibody.

Reviewer 2, Comment 1 on Discussion: “While the discussion discusses the feasibility of the strategy used in this paper and compares it to a past paper providing a balanced discussion, it does not compare it to other modification strategies used by other papers/groups. An improved literature comparison would benefit this section.”

Response: We thank the reviewer for raising that concern. We modified the Discussion section to include the following paragraph with the relevant references:

“Given the promising results obtained via protein G-mediated antibody immobilization for cell capture, other methods that aim to orient antibodies on surfaces (53) could also show significant promise in cell capture applications. Other Comparison to other antibody-based biocoatings (54-56) which lead to varying degrees of antibody orientation on surfaces could help elucidate the effect of antibody conformation on cell capture efficiency.”

Reviewer 2, Comment on Line 324, 355, 360 & 369: “Due to the confusion that remains in the field it may be best to consistently use the term ‘ECFC’ instead of ‘EPC’ as EPC can be thought of as a cell with immune cell properties.”

Response: We have removed two instances of EPC (formerly lines 324 & 355 in the introduction) but kept other instances where the context was “EPC capture” rather than discussing ECFCs. We added a reference to the consensus statement on nomenclature.

Please see Response to Reviewer 2, Comments on Lines 80, 82 for further details.

References

1. Erntell M, Myhre EB, Kronvall G. Alternative non-immune F(ab')2-mediated immunoglobulin binding to group C and G streptococci. Scand J Immunol. 1983;17(3):201-9.

2. Bjorck L, Kronvall G. Purification and some properties of streptococcal protein G, a novel IgG-binding reagent. J Immunol. 1984;133(2):969-74.

3. Akerstrom B, Nielsen E, Bjorck L. Definition of IgG- and albumin-binding regions of streptococcal protein G. J Biol Chem. 1987;262(28):13388-91.

4. Bjorck L, Kastern W, Lindahl G, Wideback K. Streptococcal protein G, expressed by streptococci or by Escherichia coli, has separate binding sites for human albumin and IgG. Mol Immunol. 1987;24(10):1113-22.

5. Erntell M, Myhre EB, Sjobring U, Bjorck L. Streptococcal protein G has affinity for both Fab- and Fc-fragments of human IgG. Mol Immunol. 1988;25(2):121-6.

6. Stone GC, Sjobring U, Bjorck L, Sjoquist J, Barber CV, Nardella FA. The Fc binding site for streptococcal protein G is in the C gamma 2-C gamma 3 interface region of IgG and is related to the sites that bind staphylococcal protein A and human rheumatoid factors. J Immunol. 1989;143(2):565-70.

7. Derrick JP, Wigley DB. Crystal structure of a streptococcal protein G domain bound to an Fab fragment. Nature. 1992;359(6397):752-4.

8. Derrick JP, Wigley DB. The third IgG-binding domain from streptococcal protein G. An analysis by X-ray crystallography of the structure alone and in a complex with Fab. J Mol Biol. 1994;243(5):906-18.

9. Kato K, Lian LY, Barsukov IL, Derrick JP, Kim H, Tanaka R, et al. Model for the complex between protein G and an antibody Fc fragment in solution. Structure. 1995;3(1):79-85.

10. Bae YM, Oh BK, Lee W, Lee WH, Choi JW. Study on orientation of immunoglobulin G on protein G layer. Biosens Bioelectron. 2005;21(1):103-10.

11. Kim ES, Shim CK, Lee JW, Park JW, Choi KY. Synergistic effect of orientation and lateral spacing of protein G on an on-chip immunoassay. Analyst. 2012;137(10):2421-30.

12. Tsai CW, Jheng SL, Chen WY, Ruaan RC. Strategy of Fc-recognizable Peptide ligand design for oriented immobilization of antibody. Anal Chem. 2014;86(6):2931-8.

13. Hoesli CA, Bashth O, Elkhodiry MA, Boulanger M, Laroche G, inventorsDual function surface for cell capture and spreading2021 2020-07-14 (USPTO), 2021-07-14 (PCT).

14. Bashth OS, Elkhodiry MA, Laroche G, Hoesli CA. Surface grafting of Fc-binding peptides as a simple platform to immobilize and identify antibodies that selectively capture circulating endothelial progenitor cells. Biomater Sci. 2020;8(19):5465-75.

15. Elkhodiry MA, Boulanger MD, Bashth O, Tanguay JF, Laroche G, Hoesli CA. Isolating and expanding endothelial progenitor cells from peripheral blood on peptide-functionalized polystyrene surfaces. Biotechnol Bioeng. 2019;116(10):2598-609.

16. Haynes CA, Norde W. Structures and Stabilities of Adsorbed Proteins. J Colloid Interf Sci. 1995;169(2):313-28.

17. Hoesli CA, Garnier A, Juneau PM, Chevallier P, Duchesne C, Laroche G. A fluorophore-tagged RGD peptide to control endothelial cell adhesion to micropatterned surfaces. Biomaterials. 2014;35(3):879-90.

18. Boivin MC, Chevallier P, Hoesli CA, Lagueux J, Bareille R, Remy M, et al. Human saphenous vein endothelial cell adhesion and expansion on micropatterned polytetrafluoroethylene. J Biomed Mater Res A. 2013;101(3):694-703.

19. Gauvreau V, Laroche G. Micropattern printing of adhesion, spreading, and migration peptides on poly(tetrafluoroethylene) films to promote endothelialization. Bioconjug Chem. 2005;16(5):1088-97.

20. Hoesli CA, Tremblay C, Juneau PM, Boulanger MD, Beland AV, Ling SD, et al. Dynamics of Endothelial Cell Responses to Laminar Shear Stress on Surfaces Functionalized with Fibronectin-Derived Peptides. Acs Biomaterials Science & Engineering. 2018;4(11):3779-91.

---

## [Editor Report · Decision Letter 1]

19 May 2022

Bioaffinity-based surface immobilization of antibodies to capture endothelial colony-forming cells

PONE-D-21-26288R1

Dear Dr. Hoesli,

We’re pleased to inform you that your manuscript has been judged scientifically suitable for publication and will be formally accepted for publication once it meets all outstanding technical requirements.

Kind regards,

Oksana Lockridge, Ph.D.

Academic Editor

PLOS ONE

---

## [Editor Report · Acceptance letter]

22 Aug 2022

PONE-D-21-26288R1 

Bioaffinity-based surface immobilization of antibodies to capture endothelial colony-forming cells 

Dear Dr. Hoesli:

I'm pleased to inform you that your manuscript has been deemed suitable for publication in PLOS ONE. Congratulations! Your manuscript is now with our production department. 

Kind regards, 

on behalf of

Dr. Oksana Lockridge 

Academic Editor

PLOS ONE